# Asynclitism and Its Ultrasonographic Rediscovery in Labor Room to Date: A Systematic Review

**DOI:** 10.3390/diagnostics12122998

**Published:** 2022-11-30

**Authors:** Antonio Malvasi, Marina Vinciguerra, Bruno Lamanna, Eliano Cascardi, Gianluca Raffaello Damiani, Giuseppe Muzzupapa, Ioannis Kosmas, Renata Beck, Maddalena Falagario, Antonella Vimercati, Ettore Cicinelli, Giuseppe Trojano, Andrea Tinelli, Gerardo Cazzato, Miriam Dellino

**Affiliations:** 1Unit of Obstetrics and Gynaecology, Department of Biomedical Sciences and Human Oncology, 70124 Bari, Italy; 2Department of Maternal and Child Health, Madonna delle Grazie Hospital, 75100 Matera, Italy; 3Fetal Medicine Research Institute, King’s College Hospital, London SE5 9RS, UK; 4Department of Medical Sciences, University of Turin, 10126 Torino, Italy; 5Pathology Unit, FPO-IRCCS Candiolo Cancer Institute, Str. Provinciale 142 km 3.95, 10060 Candiolo, Italy; 6Department of Obstetrics and Gynecology, Ioannina State General Hospital G. Chatzikosta, 45332 Ioannina, Greece; 7Department of Anesthesia and Intensive Care, Policlinico Riuniti Hospital, University of Foggia, 71122 Foggia, Italy; 8Department of Obstetric and Gynecology, Lund University, SE-221 00 Lund, Sweden; 9Department of Obstetrics and Gynecology and CERICSAL (CEntro di RIcerca Clinico SALentino), Veris Delli Ponti Hospital, 73020 Scorrano, Italy; 10Division of Experimental Endoscopic Surgery, Imaging, Technology and Minimally Invasive Therapy, Vito Fazzi Hospital, 73100 Lecce, Italy; 11Section of Pathology, Department of Emergency and Organ Transplantation (DETO), University of Bari “Aldo Moro”, 70124 Bari, Italy; 12Department of Obstetrics and Gynecology, San Paolo Hospital, 70124 Bari, Italy

**Keywords:** asynclitism, fetal head malposition, intrapartum ultrasound, vaginal examination, cesarean section, operative delivery, caput succedaneum, birth trauma, sonography

## Abstract

Asynclitism, the most feared malposition of the fetal head during labor, still represents to date an unresolved field of interest, remaining one of the most common causes of prolonged or obstructed labor, dystocia, assisted delivery, and cesarean section. Traditionally asynclitism is diagnosed by vaginal examination, which is, however, burdened by a high grade of bias. On the contrary, the recent scientific evidence highly suggests the use of intrapartum ultrasonography, which would be more accurate and reliable when compared to the vaginal examination for malposition assessment. The early detection and characterization of asynclitism by intrapartum ultrasound would become a valid tool for intrapartum evaluation. In this way, it will be possible for physicians to opt for the safest way of delivery according to an accurate definition of the fetal head position and station, avoiding unnecessary operative procedures and medication while improving fetal and maternal outcomes. This review re-evaluated the literature of the last 30 years on asynclitism, focusing on the progressive imposition of ultrasound as an intrapartum diagnostic tool. All the evidence emerging from the literature is presented and evaluated from our point of view, describing the most employed technique and considering the future implication of the progressive worldwide consolidation of asynclitism and ultrasound.

## 1. Introduction

Asynclitism, the most studied malposition of the fetal head during labor in the literature, still represents to date an unsolved field of interest, remaining one of the most common causes of prolonged or obstructed labor, dystocia, assisted delivery (AD), and cesarean section (CS) [1,2,3]. To date, asynclitism represents one of the most feared and difficult-to-manage conditions in the delivery room, which could be due to the lack of univocal and worldwide shared diagnostic and management indications. We sought to address this via a literature review. The timeline (Figure 1), describing the research trend in the last 30 years on this topic, and the table (Table 1), reporting all the papers found, lead to several and interesting consideration. Searching the word “asynclitism” on PubMed, MedLine, Scopus, and ResearchGate, we found since a total of 35 studies since the 1990s, in which asynclitism was the mean topic or appeared among the key words.

After a preliminary evaluation of all available literature, we rejected three articles, whose two case reports uncentered on asynclitism and were not relevant for our review, and one book chapter, a research form not considered for our review. All the other 32 papers were accepted regardless of the type (retrospective or prospective study, review, or letter to editor) and reassessed in light of the purpose of our review, highlighting several considerations. First of all, the definition of asynclitism as an entity is relatively recent, whose emergence has been gradual [28,30]. This is part of the progressive separation process from the atavistic and generic concept of malposition or labor dystocia, which has led to more specific characterizations, useful for the decision-making process. The overall amount of literature on asynclitism is numerically small, although it is a problem that is faced daily in delivery rooms all over the world [31]. Moreover, all the papers on asynclitism were written by select groups of researchers. This can be considered as not due to a lack of attention from the scientific community, but instead to the difficulty of standardization of this condition. Nevertheless, these groups of researchers have invented and described several technique and signs over the years, which has led to the current consolidation of asynclitism and ultrasound. The fil rouge among the several papers reported is that the more accurate the ultrasound diagnostic tools are for asynclitism, the more diversified the labor and delivery management options become [33]. However, the scientific evidence on asynclitism is limited, making it impossible to have standardized diagnostic signs and respective cutoffs or ranges of reference values. In this respect, the attempt by the Chinese research group of Chan et al. [31] to create an algorithm is very interesting [27]. They selected some of the most studied ultrasonographic signs for fetal head assessment, including for asynclitism diagnosis, and proposed their corresponding cutoffs, distinguishing between favorable and unfavorable factors, in order to predict through their combination the likelihood of successfully having an operative vaginal delivery in the case of prolonged labor [31]. Future research will surely aim to standardize intrapartum ultrasound investigations for asynclitism detection to make it easier and more usable.

## 2. Digital Examination versus Intrapartum Ultrasound

Asynclitism diagnosis has always been a challenge in the past because of several reasons. First, there is a lack of a specific characterization of this entity, which often led to it simply being labeled as fetal head malposition, without giving any further or useful information to the decision-making process [29,31]. In addiction, asynclitism diagnosis in the pre-ultrasound era was exclusively achieved via digital examination, with the fetal sagittal suture as the only landmark. However, as revealed in the literature, during labor dystocia, asynclitism often overlaps with other fetal head attitudes, which further complicates correct diagnosis by digital examination [34]. A prolonged second stage of labor asynclitism is mostly associated with caput succedaneum, which is also known as birth tumor, derived Latin etymology, meaning “swelling” [35]. In fact, it is a common benign subcutaneous edema on the fetal scalp due to the strain of an incorrect fetal head descending through birth canal, which prolongs the labor duration and amplifies the pressure of vaginal walls on the skull [36,37,38]. The swelling typically crosses cranial suture lines and the midline; thus, digital examination often does not manage to establish the persistence or not of an asynclitic attitude, which is a significant prognostic factor [39,40]. The addition of ultrasonographic evaluation has significantly improved asynclitism diagnosis, making a previously uncertain diagnosis objective and documented by means of specific and multiple easily measurable parameters [4,5]. Palpation of the ear for digital diagnosis is possible only in case of severe anterior (anterior ear) and posterior (posterior ear) asynclitism and in prolonged and neglected labors. In contrast, on average, it is difficult to palpate the ear in asynclitism [29,32]. The introduction of the use of ultrasound in the delivery room was late compared to its advent in other fields of gynecology and obstetrics. This is because ultrasound evaluation was perceived as a substitute to digital evaluation and, thus, initially frowned upon. Sherer et al., in 2002, proposed a new interpretation key, in which the ultrasound approach became a complementary tool to digital evaluation, improving its reliability and prognostic value, mostly in obstructive labor [7]. This scientific evidence was then confirmed by further studies, published in the first decade of the 2000s, which investigated the fetal head attitude assessment during labor, without specific mention of asynclitism [41,42,43,44,45]. Buchmann et al., in 2008, were the first to describe a cluster of ultrasound parameters useful for ultrasound evaluation explicitly for asynclitic head during labor [6]. Asynclitism was not yet considered an entity of its own, instead seen as belonging to a group of unfavorable fetal head attitudes, according to the wider concept of cephalopelvic disproportion. From that moment on, all research groups on asynclitism focused not only on digital diagnosis but also on ultrasound evaluation, defining it from periodically [29,31].

## 3. Asynclitism: A New Old Entity

Asynclitism occurs when the fetal head enters the obstetric pelvis at an abnormal angle, such that the sagittal suture is not directly at the median sagittal plane of the maternal pelvis. Anterior asynclitism occurs when a posterior twisting of the fetal head makes the sagittal suture close to the sacrum with the anterior parietal bone as the presenting part. Similarly, in posterior asynclitism, an anterior twisting of the fetal head makes the sagittal suture close to the pubis with the posterior parietal bone as the leading part [2]. There are three degrees of asynclitism; the first (mild) is physiological asynclitism with 15 mm deviation of the sagittal suture from the midline axis, the second (moderate) is when the sagittal suture approaches the pubic joint or sacral promontory, and the third (severe) is when the suture extends beyond those two [32]. Asynclitism is caused by anomalies of the birth canal (i.e., android pelvis, cephalopelvic disproportion, myomas, or other soft-tissue structure) and an abnormal descent of the leading part through the birth canal [1,3,46]. An incomplete internal rotation of the fetal head during labor can lead to fetal head malposition, such as persistent occiput posterior position (POPP) (5–7% of labors) or occiput transverse position (OTP), which can also lead to asynclitism, increasing the rate of labor dystocia and maternal/fetal complications [3,8,12]. With an early diagnosis of fetal head malposition, the option of an operative vaginal delivery (OVD) might be contemplated by some, while others prefer to go the way of a cesarean section to avoid the potential risks associated with OVD [23]. Blayney et al., in 1989 [34], first reported the successful use of vacuum extractor (VE) rotation in the case of early diagnosis of asynclitism. Malvasi et al. (2011) demonstrated that asynclitism is a consequence of intrapartum cephalopelvic disproportion (CPD), resulting in a CS due to poor labor progress despite adequate uterine contractions [11]. Therefore, with an earlier diagnosis of CPD, intrapartum management is expected to be better. According to Buchmann et al. (2008), CPD can be predicted by the fetal sagittal and lambdoid sutures overlap, and it is classified into three grades: grade 0 (skull bones separated), grade 1 (touching bones), and grade 2 (overlapping bones). The positive predictive value of sagittal suture overlap is 47.1% with grade 2 and 34.5% with grade 1 CPD. On the other hand, the positive predictive values of lamboid suture overlap are 25.2% and 23.4% for grade 2 and grade 1, CPD respectively. The absence of a sagittal or a lamboid suture overlap had 87.2% (211/242) and 90.9% negative predictive values for CPD, respectively [6]. According to the literature, compared to the vaginal digital examination (VDE), ultrasound (US) is associated with an earlier, more reliable, and more accurate diagnosis of asynclitism, especially in misleading cases, such as caput succedaneum, which is often observed in labor dystocia [24]. Malvasi et al. (2011) demonstrated that intrapartum transabdominal sonography (ITAS) has a lower margin of error for fetal head assessment and asynclitism detection compared to VDE, especially during the first stage of labor when the cervix is not dilated enough to allow the examiner to assess the sagittal sutures and fontanelles [1]. In 2012, Ghi et al. described the intrapartum use of translabial ultrasound at 8 cm dilatation to diagnose asynclitism. A three-dimensional US reconstruction of the transverse view of an asynclitic head was obtained by cutting the volume with an oblique line rather than perpendicular to the pubis, as in the case of a transverse synclitic head [2]. Malvasi et al. characterized the ITAS signs of anterior and posterior asynclitism [1]. Anterior asynclitism is represented by the visible anterior orbits (“anterior squint sign” or “Malvasi’s sign 1”) (Figure 2A,B; Figure 3A,B), the north thalamus or cerebellum sunset (“Malvasi’s sign 2 and 3”) (Figure 4), or when the midline is <9 o’clock and >3 o’clock. On the other hand, posterior asynclitism occurs in the case of a “posterior squint sign”, “south thalamus and cerebellum sunset”, and midline >9 o’clock and <3 o’clock [1,12].

Malvasi et al. successfully used these signs to diagnose and characterize transverse asynclitism on a sample of 150 women, which helped in determining the optimal time and technique for operative delivery with minimal complications [12]. Ghi et al. reported a combined use of a suprapubic and a transperineal US scan to diagnose posterior asynclitism in cases where a VDE was considered misleading (slow progression of labor with fetal head at station 0 and a large caput succedaneum). A CS was considered as the safest option for posterior asynclitism [22].

In 2015, Ghi et al. [15] and our group [17] described so-called lateral asynclitism, which causes obstructed labor at the first stage. Ghi et al. [15] stated that it occurs when the head is laterally oriented by 90° in relation to the sagittal plane and descends in a mento-cervical position, whereby the sagittal suture can be palpated laterally, and the two fontanelles cannot be detected, making the diagnosis only possible by US. Malvasi et al. [17] specified that, in lateral asynclitism, the suboccipital–bregmatic diameter is parallel to the anterior–posterior diameter of the pelvis, and it is an indication for operative delivery, despite the bitrochanteric diameter aligned to the pelvic ischiatic diameter increasing the risk of severe dystocia. Ghi et al. [15], instead, opted for CS in all cases described. In 2016, Malvasi et al. demonstrated that the severity of asynclitism is linked to the need for operative delivery and likelihood of obstetric complications, such as severe dystocia. They performed a transabdominal transversal 2D US and described two useful and simple signs to distinguish between a light or a marked asynclitism. In light asynclitism, the fetal head is partially twisted, while the nose and one orbit are visible, which is known as “the squint sign with or without nose”. However, when the head is completely flexed on the homolateral shoulder and/or column, the nose and orbits are not visible, which is diagnostic of a marked asynclitism [18,20,32].

Recently Habek et al. focused on the “hard-to-handle” association between asynclitism and POPP, describing a case with posterior low-lying focal increta placentomegaly ending up with dystocia [28]. In a prospective observational study performed on a large number of nulliparous women, Malvasi et al. found that, in all cases of asynclitism, either OPP rotating in an anterior position or POPP increased the likelihood of a prolonged second stage of labor, which was always >180 min. Among all asynclitism and POPP cases, 73% ended with operative delivery (54% CS and 19% VE), with a low angle of progression (AoP < 100°) and an unfavorable pubic arch angle (PAA), as predicted by intrapartum US [26]. Chan et al., using a prediction algorithm based on clinical and sonographic parameters, defined three groups of patients according to “traffic lights”, green/yellow/red, directly correlated with the likelihood of each method of delivery. The red group, which had the highest rate of operative delivery, had more than one unfavorable US parameter, e.g., OPP, “head-down” direction, AoP at rest <120°, head–perineum distance (HPD) at rest ≥40 mm, and δHPD (the difference between HPD during lowest and maximum contraction) negative or ≤2 mm. Instead, the green group, which had the highest rate of SVD, had only favorable US signs, e.g., “head-up” direction and HPD at rest ≤25 mm [31]. Hinkson et al., in 2021 [27], introduced a novel method for real-time intrapartum US with simultaneous use of Kielland’s rotational forceps, aimed at reducing the CS rate. They managed to successfully achieve rotation of the fetal head to the occiput anterior position in all 32 women, leading to an uncomplicated OVD.

US malposition assessment is crucial for the intrapartum decision-making process; therefore, as suggested by Malvasi et al., it is useful to collect and store photos in medical records for possible forensic administrative issues in the future [20].

## 4. Operative Vaginal Delivery: Forceps and Vacuum

The lack of progression of the fetal head during labor for at least 2–3 h from full dilatation despite adequate myometrial activity necessitates an operative delivery, which consists of two options: OVD or CS. Back in 1987, Compton et al. demonstrated that X-ray pelvimetry could provide information about the real station and angle of the fetal head, thus helping to manage pelvic or soft-tissue dystocia while avoiding unnecessary forceps trials in favor of CS [46]. Soon after, in 1989, Blayney reported five cases of prolonged labor all caused by an asynclitic attitude of the fetal head. A digital examination was performed to diagnose malposition and to guide vacuum application, which successfully resulted in uncomplicated vaginal delivery. All the women were multiparous, which could be a favorable factor for uncomplicated asynclitism resolution using a suction cup [34]. In 1996, Bofill et al., in a randomized study on 637 women, compared the effectiveness of the obstetric forceps (n = 315) and the M-cup VE (n = 322). They found that M-cup was a more effective and safer instrumental delivery tool than the forceps, having efficacy rates of 94% and 92%, respectively, while causing fewer maternal genital tract lacerations and episiotomies, despite the same amount of blood loss. The M-cup was easier and quicker to apply, but was associated with more diagnosed cephalhematomas. However, there was no difference in the incidence of neonatal hyperbilirubinemia or need for phototherapy [4]. A year later, Bofill et al., in a randomized controlled trial of 322 women at ≥34 weeks gestation, compared continuous (n = 164) and intermittent (n = 158) VE application. They found that cephalhematomas incidence was not influenced by the method of VE application, but by other factors, e.g., station at point of application, degree of asynclitism, and increasing interval between VE application and delivery time. Only 28% of the neonates presented with cephalhematoma when time from vacuum application to delivery exceeded 5 min [5]. On the other hand, in a prospective observational study, Hinkson et al. reported their successful experience with Kielland’s rotational forceps in 32 women during a prolonged second stage of labor. Fetal head malposition (occiput posterior or transverse) was diagnosed by US. In the case of instrument-assisted vaginal birth, it is important that the station of the presented part is placed between the ischial spines (i.e., the head must be engaged); however, if the degree of asynclitism is marked and this can be evaluated by ultrasound, the application of vacuum extraction is not indicated, as its application on the parietal bone can cause trauma to the venous sinuses resulting from traction and fetal intracranial hemorrhages. In the literature, it is even reported that rotational forceps would be preferable in these cases. In particular, in a prospective observational study, Hinkson et al. reported their successful experience with Kielland’s rotational forceps in 32 women during a prolonged second stage of labor [47,48,49]. There were no cases of difficult or repeated applications, slippage of the blades, or incorrect rotation of the fetal head. There were no vaginal or cervical tears, except for one case of third-degree perineal tear. There was one neonate with mild hyperbilirubinemia and one with small cephalhematoma, which was managed conservatively [27].

### 4.1. Labor Analgesia

Labor analgesia is reported to be the most effective technique for pain control during labor [16,50,51]. Bofill, in 1996, compared the effectiveness of forceps and VE in terms of the need for analgesia during labor. The use of epidural, saddle block, and local infiltration was similar. Instead, they found that the use of forceps significantly increased the need for pudendal block anesthesia or more than one type of anesthesia, such as pudendal and local, being perceived as a more painful procedure by the patient [4]. In 2011, Malvasi et al. compared the possible influence of combined spinal–epidural analgesia (CSE) or no neuraxial labor analgesia (NLA) at all on asynclitism. They found that applying CSE early in labor and using low doses during labor did not increase the rate of obstructed labors because asynclitism is not caused by medication, but by CPD [11]. Beck et al.’s review in 2019 reported that NLA is unsuccessful in dystocic prolonged labor, but does not increase the rate of OVD and CS, as also confirm trials of Shen et al. and Wang et al., because modern NLA uses low doses of local anesthetics [25,52,53]. In agreement with their previous study, Malvasi et al., in 2020, found that a prolonged NLA in the case of POPP and asynclitism prolonged the second stage of labor, causing maternal complications, such as anatomical modifications of the lower uterine segment, e.g., bulging. Therefore, prolonged NLA is no longer useful in asynclitic labor dystocia and should be discontinued [4].

### 4.2. US in the First Stage of Labor

Intrapartum US can be a useful tool for fetal head assessment, mostly in the case of slow progress or arrest of the leading part during labor dystocia. The International Society of Ultrasound in Obstetrics and Gynecology (ISUOG) recommends a transabdominal US scan to define the fetal head position, along with a transperineal US for station detection. US evaluation during the first stage of labor has been documented and accredited by several researchers, as mentioned above. Ghi et al. reported a case of both anterior and posterior asynclitism diagnosed and managed trough intrapartum US during the first stage of labor. In the first case, a translabial US scan (Figure 5) confirmed the suspicious of anterior asynclitism aroused by VDE, using a 3D technique to obtain a transverse view of the fetal head. Ghi et al. also remarked the importance of obtaining an axial view of the fetal head on transperineal US to characterize posterior asynclitism in the first stage of labor, so as to avoid an unnecessary and risky OVD in favor of CS [2,22]. 

Moreover, according to Ghi et al., the diagnosis of lateral asynclitism, which requires CS, has to be achieved using transabdominal US during the first stage of labor in order for a CS to be performed in a timely manner. Ghi et al. stated that the simultaneous vision of the fetal profile and the chest with an apical four-chamber view along the axial plane is pathognomonic for lateral asynclitism diagnosis. Malvasi et al. compared the diagnostic performance of intrapartum US and VDE in detecting asynclitism and an occiput-transverse head position during labor dystocia. They used a suprapubic scan along the transverse plane and a transperineal scan along the transverse and sagittal planes to identify the fetal head position. The fetal landmarks of reference were the orbits, midline, thalamus, and cerebellum. They found that US is more efficient than VDE during both the first and second stages of labor [12]. An optimal tool which can be regularly used during the first stage of labor is the “traffic-light” algorithm model proposed by Chan et al., which was described above, during triage in order to avoid risky or useless attempts of vaginal [31] delivery.

### 4.3. US in the Second Stage of Labor

A prolonged second stage of labor is often associated with OVD and CS. Fetal head malposition and malrotation, such as POPP, mostly plus asynclitism, are the main causes of obstructed labor. This increases the incidence of maternal and neonatal morbidities [26]. Because of the frequent presence of head molding and caput succedaneum during labor dystocia, VDE may be problematic and misleading for the diagnosis of asynclitism, thus necessitating confirmation with US [16]. This agrees with the findings of Hung et al. in their 1 year prospective study, which evidenced that US accuracy is higher than that of VDE mostly in the non-occiput anterior position of the fetal head. They performed a transperineal scan when the leading part was deeply engaged and identified asynclitism according to the “asynclitic midline sign”, consisting of a fetal head midline not equidistant from the biparietal bones. They found that asynclitic heads presented a larger HPD during pushing, but a similar HPD at rest compared to synclitic heads. US allowed detecting more cases of asynclitism during the second stage of labor than expected [30]. US scanning during the second stage of labor also allows the detection of lateral asynclitism, which is characterized by the suboccipital–bregmatic diameter of the fetal head being parallel to the anterior–posterior diameter of the maternal pelvis. A transabdominal scan can identify the exact fetal spine position, anterior or posterior, whereas the lateralization of the squint sign can be seen directly using a transvaginal scan or indirectly using a transabdominal view of the asymmetrical fetal profile on a longitudinal plane [17,52]. Moreover, in the case of OVD, intrapartum US can be of guidance for the correct application of forceps, confirming the correct rotation of the fetal head and, thus, improving the success rate of uneventful instrumental delivery, as reported by Hinkson et al. [27]. Lastly, as observed by Malvasi et al., in the case of a prolonged second stage of labor dystocia and when NLA is used, US landmarks, such as AoP and PAA, can allow timely detection of fetus malposition, predicting the type of delivery and avoiding unnecessary prolongation of labor and NLA [25,26].

### 4.4. Complications

In a prospective randomized controlled trial of vacuum-assisted delivery, Bofill et al. concluded that fetal head asynclitism was the only prenatal factor that could be related to neonatal cephalhematoma formation, one of the most severe obstetric complications [5]. Moreover, Hinkson et al., in an observational study, referred to a case of neonatal cephalhematoma after rotational forceps delivery, which resolved spontaneously, without neurological complications [27]. Lastly, Kwan et al. reported a case of severe asynclitism, which led to obstruction and shoulder dystocia ending up with fetal death during delivery [33]. Malvasi et al. described the mechanism of subaponeurotic hemorrhages caused by asynclitism and operative delivery. Asynclitism is easily diagnosed in dead fetuses and newborns by the localization of the region of periosteal blood congestion (RPC). The RPC shifts on the surface of the right or left parietal bone, occupying a large area, and the leading point also shifts accordingly. Therefore, they studied if there was a connection between RPC and tentorium cerebelli (TC) ruptures. They observed that, when RPC was central, there were bilateral ruptures, whereas, when RPC was asymmetric due to asynclitism, the ruptures were predominantly on one side. Additionally, when RPC was bilateral, the damage occurred on the opposite side. The degree of asynclitism directly correlated with the degree of injury. They concluded that dynamic monitoring of the state of asynclitism during labor by intrapartum US is important in order to predict and prevent complications and birth trauma [32]. Vlasyuk et al. demonstrated that, while VE reduces perineal injuries, it can have adverse effects on the fetus. The negative pressure caused by the cup attachment and traction to the fetal head could induce circulatory disorders, subcutaneous hemorrhages, and thus, hypoxia and acidosis in the underlying tissues. In the case of asynclitism, VE could enhance it, worsening birth trauma. On the contrary, forceps does not cause hemorrhages, but has to be used only with a movable lock in order to have full control over the tongs, because overlapping can cause significant injuries, such as fractures of parietal bones [32].

## 5. Expert Opinion and Conclusions

This narrative review had the aim of highlighting the utility of intrapartum ultrasound for a correct fetal head attitude, mostly in the case of obstructive labor or labor dystocia. Its addiction to vaginal digital examination can significantly improve the accuracy of diagnosis. Our experience agrees with both the literature and the most recent international guidelines, i.e., that intrapartum US should be mandatory for malposition and asynclitism assessment. Intrapartum US is simple and intuitive. Its diagnostic accuracy ensures a reliable prediction of the way of delivery and allows avoiding unnecessary and risky procedures. Moreover, what emerges from the literature is the association of asynclitism with other fetal head attitudes, such as a posterior occiput position and persistent occiput position, which can be defined correctly only by ultrasound investigation before a decision on the delivery method. The application of ultrasound during labor is leading to the renaissance of abandoned techniques, such as rotational forceps, whose application should be safer under ultrasound guidance, avoiding the risk of cephalohematoma related to suction cup. As a matter of fact, expert opinions and guidelines promote intrapartum ultrasound evaluation in the case of suspected asynclitism according to digital examination before applying a vacuum extractor. This should significantly reduce the risk of birth trauma caused by improper or incorrect suction cup application. Asynclitism is also often associated with a transverse posterior occiput position. In this setting, the ultrasound detection of the head position during prolonged labor in a patient under neuraxial analgesia can lead to diagnosis of the type of dystocia and program the most operative birth, suspending further analgesic drug somministration. Lastly, intrapartum US represents an objective and recordable support tool in a physician’s decision making, avoiding any medico-legal aftermath.

## Figures and Tables

**Figure 1 diagnostics-12-02998-f001:**
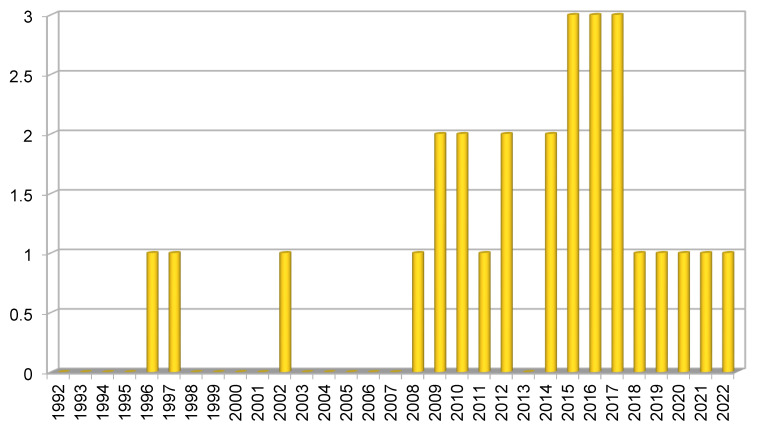
Timeline histogram of all articles on asynclitism published in the last 30 years. As discussed in the paper, the attention of the scientific community toward asynclitism has increased mostly in the last 20 years, although the number of all the papers available in the literature on this topic is still unsatisfactory to definitely standardize asynclitism diagnosis and management.

**Figure 2 diagnostics-12-02998-f002:**
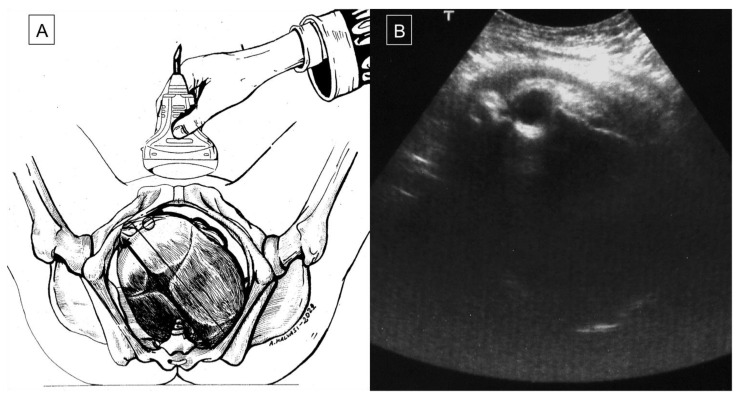
During a prolonged second stage of labor, the operator assesses the fetal head using a suprapubic transabdominal scan, orienting the probe transversely (**A**); the occiput position is left posterior, and the right orbit clearly visible in the upper left quadrant of the screen is the “anterior squint sign”, which enables diagnosis of an anterior asynclitism fetal head attitude (**B**).

**Figure 3 diagnostics-12-02998-f003:**
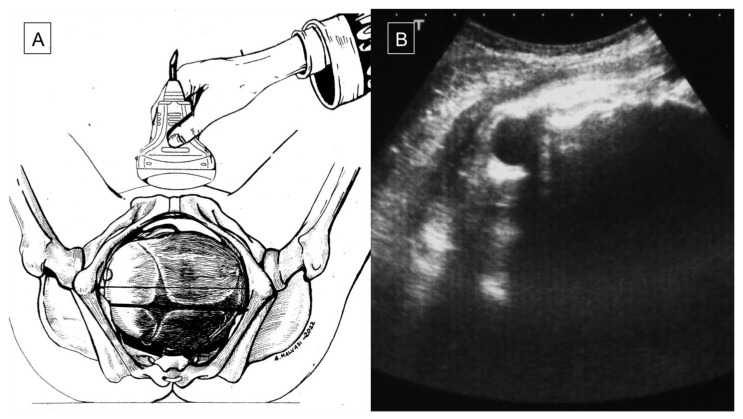
(**A**,**B**) Suprapubic transabdominal ultrasound exam during labor dystocia: a transverse section of the fetal head through a probe oriented transversely on the lower abdomen allows detecting the fetal head attitude; the midline is perpendicular to the maternal pelvis sagittal plane in a transverse occiput position (**A**), whereas the midline appears shifted down and the anterior or right parietal bone is predominant with respect to the contralateral bone on the transverse scan; thus, the anterior or right orbit is visible as an “anterior squint sign”, which leads to a diagnosis of left transverse occiput position plus anterior asynclitism (**B**).

**Figure 4 diagnostics-12-02998-f004:**
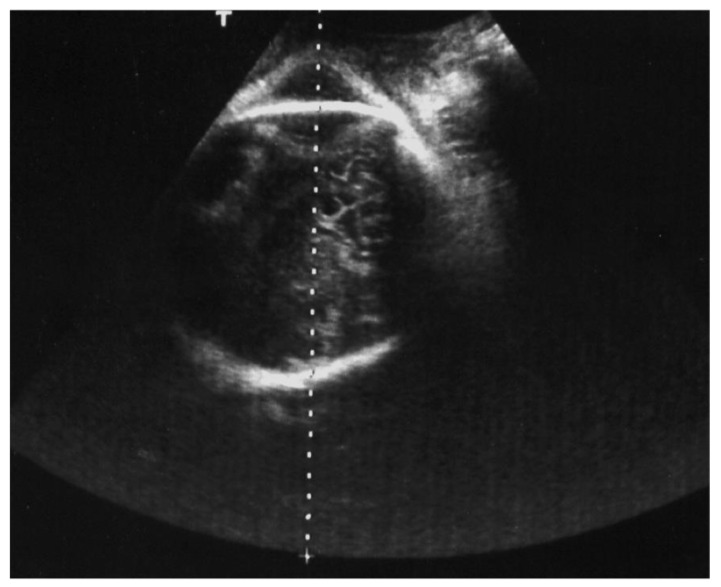
Suprapubic transabdominal sonograph for fetal head assessment; the transverse scan reveals that the occiput position is left anterior, and the cerebellum appears on the upper right quadrant of the screen, the so-called “north cerebellum sign”, typical of anterior asynclitism fetal head attitude.

**Figure 5 diagnostics-12-02998-f005:**
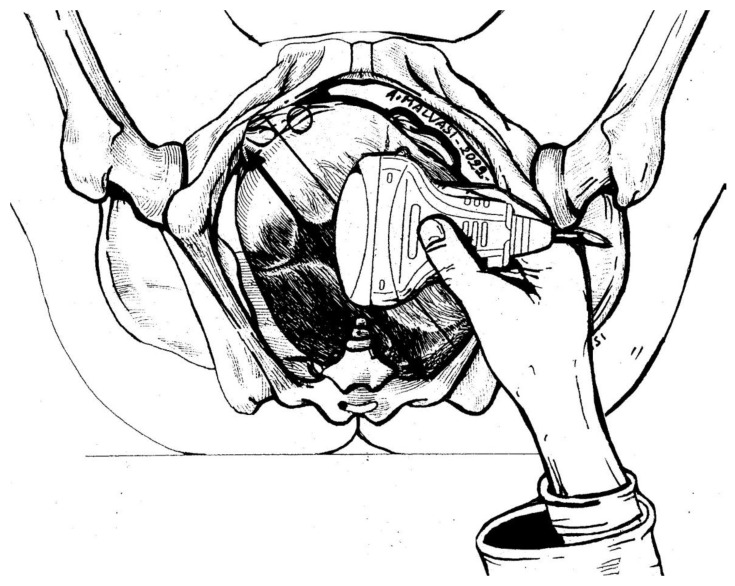
The translabial sonography technique was introduced in the labor ward later than transabdominal intrapartum ultrasound. The operator must place the linear probe between the labia majora of the parturient in different orientations according to the several signs considered. In this picture, the probe is at the sagittal plane of the pelvis, which is a landmark to compare the fetal head midline orientation in order to discriminate the fetal head attitude. In this picture, the occiput position is posterior left plus anterior asynclitism. Using a 3D linear probe, it is possible to obtain from this scan further planes of reconstruction to better assess or confirm fetal head malposition or malpresentation.

**Table 1 diagnostics-12-02998-t001:** All the articles available in literature in the last 30 years on asynclitism with a short description of their main characteristics and relevant results. Abbreviations: 4CW, four-chamber view; AoP, angle of progression; ASC, asynclitism; CPD, cephalopelvic disproportion; CS, cesarean section; DE, digital vaginal examination; FP, forceps; HD, head direction; HPD, head progression distance; HR, head rotation; IPU, intrapartum ultrasound; US, ultrasound; LNA, labor neuraxial analgesia; N°, number of papers mentioned in the review; OSA, occiput–spine angle; PAA, pubic arch angle; POPP, persistent occiput posterior position; PS, prospective cross-sectional study; RCT, randomized control trial; TA, transabdominal suprapubic ultrasound scan; TL, translabial ultrasound scan; TP, transverse position; TTC, tears of tentorium cerebellum; VE, vacuum extractor; YR, year of publication; vs., versus.

Papers ID	YR	Type of Articles	N°	DE (Yes/No)	IPU (Yes/No)	US Signs	Relevant/*p*-Value Positive Results
Bofill et al. [4]	1996	RCT	-	yes	no	-	M-cup VE vs. FP (+fast, −episio/lacerations, +cephalhematomas)
Bofill et al. [5]	1997	RCT	-	yes	no	-	VE and cephalhematomas (+asynclitism degree, +VE delivery time)
Buchmann et al. [6]	2008	PS	-	yes	yes (TA)	-	Predictors of CPD (sagittal suture overlap, cervical dilatation, level of head, caput succedaneum, active labor duration, birth weight)
Sherer et al. [7]	2002	RCT	-	yes	yes (TA)	Occiput position Intracranial midline	TA-IPU vs. DE alone (+accurate)
Hanson [8]	2009	Review	18	-	-	-	-
Akmal et al. [9]	2009	Review	9	yes	yes (TA)	Squint sign Angle of tilt	-
Malvasi et al. [1]	2010	Letter to Editor	5	yes	yes (TA)	Squint sign Angle of tilt	-
Barber et al. [10]	2010	Review	40	yes	yes (TA)	Angle of progression	-
Malvasi et al. [11]	2011	RCT	-	yes	yes (TA)	Squint sign Sunset thalamus Cerebellum signs	-
Malvasi et al. [12]	2012	RCT	-	yes	yes (TA, TL)	Occiput position Intracranial midline	IPU vs. DE (+accurate in TP and TP + ASC)
Ghi et al. [2]	2012	Case report	-	yes	yes (TA, 3D)	Intracranial midline Angle of tilt	-
Malvasi et al. [3]	2014	Review	33	yes	yes (TA, TL)	-	-
Vlasiuk et al. [13]	2014	Retrospective study	-	yes	-	-	Degree of ASC correlates with degree of birth trauma of the skull
Vlasyuk et al. [14]	2014	Retrospective study	-	yes	-	-	Degree of ASC correlates with degree of birth trauma of the brain
Ghi et al. [15]	2015	Case series (n°5)	-	yes	yes (TA, TL)	TA axial view (chest + 4cw + profile)	Lateral ASC (+dystocic labor, +arrest dialatation/>4 h + urgent CS)
Malvasi et al. [16]	2015	Letter to Editor	12	-	yes (TL, 3D)	Squint sign ant/post/lateral (left or right)	-
Malvasi et al. [17]	2015	Letter to Editor	5	yes	yes (TA, TL)	Asymmetrical profile	-
Malvasi et al. [18]	2016	Letter to Editor	5	yes	yes (TA)	Transverse view visible: Orbit + nose only orbit	-
Malvasi et al. [19]	2016	Review	30	Yes	yes (TA, TL)	Head assessment, Occiput position, AoP, HPD, HD, HR, PAA, OSA, ASC signs (squint sign thalamus/cerebellum sunset), midline shift.	-
Malvasi et al. [20]	2016	PS	-	yes	yes (TA, TL)	Head assessment, ASC signs	Smartphone as low-cost, reliable legal proof in labor dystocia
Bellussi et al. [21]	2017	Review	69	yes	yes (TA, TL)	OSA, ASC signs, midline shift	-
Ghi et al. [22]	2017	Case Reports	-	yes	yes (TL)	Perpendicular skull section	IPU-TL vs. DE (+accurate in posterior ASC)
Malvasi et al. [23]	2017	Letter to Editor	15	yes	yes TA, TL	Head assessment, ASC signs	IPU vs. DE (+accurate diagnosis in POPP + ASC, +safe manual rotations)
Malvasi et al. [24]	2018	Letter to Editor	5	yes	yes (TA, TL)	ASC signs, midline shift	-
Beck et al. [25]	2019	Review	52	yes	yes (TA, TL)	Head assessment ASC signs	IPU (earlier diagnosis of labor dystocia, −useless NLA in dystocic labor)
Malvasi et al. [26]	2020	PS	-	yes	yes (TA, TL)	AoP, PAA, CS	IPU predicts the type of delivery
Hinkson et al. [27]	2021	PS	-	yes	yes (TA)	Occiput position, Intracranial midline direction	IPU allows easier and safer forceps application in all cases studied
Habek et al. [28]	2021	Case Report	-	yes	-	-	-
Gimovsky [29]	2021	Review	52	yes	yes (TA, TL)	Head assessment ASC signs	IPU aids in decision making
Hung et al. [30]	2021	PS	-	yes	yes (TA, TL)	Occiput position, Intracranial midline direction, HPD	ASC (prevalence 15%, +BMI, +in non occiput anterior, >HPD at pushing, +VE)
Chan et al. [31]	2021	Review	61	yes	yes (TA, TL)	Occiput position, HD, AoP, HPD, δ-HPD	Algorithm (IPU correlation with management)
Vlasyuk et al. [32]	2022	Review	37	Yes	yes (TA, TL)	Head assessments, ASC signs	ASC degree (+trauma, typically one side)

## Data Availability

Not applicable.

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
