# Peer review of "Asynclitism and Its Ultrasonographic Rediscovery in Labor Room to Date: A Systematic Review"

_diagnostics, 2022, doi:10.3390/diagnostics12122998_

Round 1
Reviewer 1 Report (Previous Reviewer 1)
Articles "Asynclitism and its ultrasonographic rediscovery in labor room to date: an expert review" in this present revised form with all accepted review suggestions, my comment: accept.
Author Response
Thank you for you revision.
Reviewer 2 Report (Previous Reviewer 2)
1. There is a typo in the abstract in the first sentence "u solved"
2. Another typo line 43 of manuscript "describign"
3. I completely agree that this is a review by experts in the field but it is unacceptable that out of 57 bibliographic titles 17 are self-cited by the first author and 5 of the bibliographic titles by 3 other authors in this paper. Moreover if it is a review of experts in the field of the 15 authors I find publications as first author on this topic only 4 of the authors.
4. Not very clear idea in line 95 "standardize a so undefined"
5. In the digital examination there is no mention of palpation and identification of the fetal ears, which sometimes favours the diagnosis of asynclitism. This is an important point in the diagnosis of asynclitism and the management of labour.
6. In the case of instrument-assisted vaginal birth, it is necessary to clearly state in which situations and at which stations you think the forceps or vacuum should be applied, because modern indications have nuanced the use of both methods.
Author Response
Please note that we have done a major overhaul of our manuscript and are grateful for the reviewer's suggestions. Further changes were made as required.

Round 2
Reviewer 2 Report (Previous Reviewer 2)
I agree with what you suggested about self-citation, but this should also be put to the editor to decide if it contravenes the principles of the journal and the MDPI group.
In my view once this contradiction has been resolved and decided the article represents a scientific work that deserves to be published.
Congratulations and good luck for the future!
Best regards!
This manuscript is a resubmission of an earlier submission. The following is a list of the peer review reports and author responses from that submission.
Round 1
Reviewer 1 Report
This received paper for review under the title Asynclitism and its ultrasonographic rediscovery in labor room to date: an expert review", is a review article dominated by own experiences on the mentioned topic.Thus, the paper is full of self-citations that are self-suggestive in some parts of the article, because a priori it still rejects digital detection of intrapartum aberrations in most maternity hospitals around the world. Thus, other clinical experiences show that experiential digital evaluation is superior or at least identical to ultrasound findings, however, it is not possible to differentiate the entire birth canal by palpation and assess the progress of labor after successful manual rotation.view references: (Habek D, Marton I, Prka M, Luetić A. Manual rotation in cases of the intrapartal arrest of fetal head. Eur J Obstet Gynecol Reprod Biol 2017;219:66-67. doi: 10.1016/j.ejogrb. and other ) The work is certainly worthy of attention in the world of science and obstetric controversies, but manual skills are still used in most maternity hospitals around the world, so digital evaluation is the only one, so it should not be ignored. Therefore, in a review article, other research that refutes or confirms the ultrasound, digital or combined method of intrapartum evaluation should be proven in a tabular presentation. Thus, the paper will have the characteristics of a review paper with an adequate discussion. In this form, this characteristic is absent, but autosuggestive throughout the text.
Reviewer 2 Report
This "expert review" is mainly a review of previously published work of the main author. Almost a half of the citations are articles belonging to the main author. Therefore, I do not consider it brings significant new information as opposed to previously published work.
1. The introduction is way too long; the authors intricate information regarding the different degrees of asynclitism and their characteristics, while also briefly describing how ultrasound comes into hand. Still, this part should be shortened and a separate chapter should be provided regarding particularities of asynclitism and regarding the utility of ultrasound.
2. The review lacks a clearly stated objective.
3. The applications of ultrasound are limited to two small subchapters.
4. The entire review contains only 29 bibliographic references and 12 of them belong to the first author, Malvasi A.
Reviewer 3 Report
This review describes very well the current challenges and state of knowledge of asynclitism at birth. However, I would ask the authors to add pictures when describing the malposition so that the descriptions here are more clearly presented to the reader. Likewise, the authors also describe the use of ultrasound during birth for malpositioning. Here, too, it would be very good if the authors would include corresponding ultrasound images, which would enable the reader to better follow the steps described and also to understand the respective setting sonographically. This would significantly improve the manuscript and make it more interesting for the reader.